# Grounding Language to Entities for Generalization in Reinforcement Learning

## Abstract

In this paper, we consider the problem of leveraging textual descriptions to improve generalization of control policies to new scenarios. Unlike prior work in this space, we do not assume access to any form of prior knowledge connecting text and state observations, and learn both symbol grounding and control policy simultaneously. This is challenging due to a lack of concrete supervision, and incorrect groundings can result in worse performance than policies that do not use the text at all. We develop a new model, EMMA (Entity Mapper with Multi-modal Attention) which uses a multi-modal entity-conditioned attention module that allows for selective focus over relevant sentences in the manual for each entity in the environment. EMMA is end-to-end differentiable and can learn a latent grounding of entities and dynamics from text to observations using environment rewards as the only source of supervision. To empirically test our model, we design a new framework of 1320 games and collect text manuals with free-form natural language via crowd-sourcing. We demonstrate that EMMA achieves successful zero-shot generalization to unseen games with new dynamics, obtaining significantly higher rewards compared to multiple baselines. The grounding acquired by EMMA is also robust to noisy descriptions and linguistic variation.[1]

## 1 Introduction

Interactive game environments are useful for developing agents that learn grounded representations of language for autonomous decision making (Golland et al., 2010; Andreas & Klein, 2015; Bahdanau et al., 2018). The key objective in these learning setups is for the agent to utilize feedback from the environment to acquire linguistic representations (e.g. word vectors) that are optimized for the task. Figure 1 provides an example of such a setting, where the meaning of the word *fleeing* in the context is to "move away", which is captured by the movements of that particular entity (*wizard*).

Learning a useful grounding of concepts can also help agents navigate new environments with previously unseen entities or dynamics. Recent research has explored this approach by grounding language descriptions to the transition and reward dynamics of an environment (Narasimhan et al., 2018; Zhong et al., 2020). While these methods demonstrate successful transfer to new settings, they require manual specification of some minimal grounding before the agent can learn (e.g. a ground-truth mapping between individual entities and their textual symbols).

In this paper, we propose a model to learn an effective grounding for entities and dynamics without requiring any prior mapping between text and state observations, using only scalar reward signals from the environment. To achieve this, there are two key inferences for an agent to make — (1) figure out which facts refer to which entities, and (2) understand what the facts mean to guide its decision making. To this end, we develop a new model called EMMA (Entity Mapper with Multimodal Attention), which simultaneously learns to select relevant sentences in the manual for each entity in the game as well as incorporate the corresponding text description into its control policy. This is done using a multi-modal attention mechanism which uses entity representations as queries to attend to specific tokens in the manual text. EMMA then generates a text-conditioned representation which is processed further by a deep neural network to generate a policy. We train the entire model in a multi-task fashion using reinforcement learning to maximize task returns.

---

[1] Code and data are available at `https://www.dropbox.com/s/fnprjrfekbnxxru/code_data.zip?raw=1`.

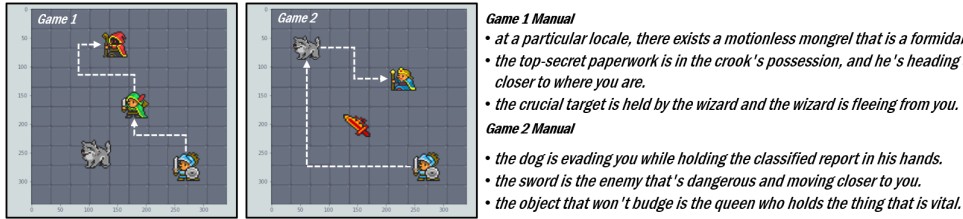

Figure 1: Two games from our multitask domain *Messenger* where the agent must obtain the message and delivers it to the goal (white dotted lines). The same entities may have different roles in different games which are revealed by the text descriptions.

To empirically validate our approach, we develop a new multi-task framework containing 1320 games with varying dynamics, where the agent is provided with a text manual in English for each individual game. The manuals contain descriptions of the entities and world dynamics obtained through crowdsourced human writers. The games are designed such that two environments may be identical except for the reward function and terminal states. This approach makes it imperative for the agent to extract the correct information from the text in order to succeed on each game.

Our experiments demonstrate EMMA is able to outperform three types of baselines (language-agnostic, attention-ablated, and Bayesian attention) with a win rate almost $40\%$ higher on training tasks. More importantly, the learned grounding helps our agent generalize well to previously unseen games without any further training (i.e. a zero-shot test), achieving up to a $79\%$ win rate. Our model is also robust to noise and linguistic variation in the manuals. For instance, when provided an additional distractor description, EMMA still achieves a win-rate of $75\%$ on unseen games.

## 2 RELATED WORK

**Grounding for instruction following**   Grounding natural language to policies has been explored in the context of instruction following in tasks like navigation (Chen & Mooney, 2011; Hermann et al., 2017; Fried et al., 2018; Wang et al., 2019; Daniele et al., 2017; Misra et al., 2017; Janner et al., 2018), games (Golland et al., 2010; Reckman et al., 2010; Andreas & Klein, 2015; Bahdanau et al., 2018; Küttler et al., 2020) or robotic control (Walter et al., 2013; Hemachandra et al., 2014; Blukis et al., 2019) (see Luketina et al. (2019) and Tellex et al. (2020) for more detailed surveys). Recent work has explored several methods for enabling generalization in instruction following, including environmental variations (Hill et al., 2020a), memory structures (Hill et al., 2020c) and pre-trained language models (Hill et al., 2020b). In a slightly different setting, Co-Reyes et al. (2018) use incremental guidance, where the text input is provided online, conditioned on the agent's progress in the environment. Andreas et al. (2017) developed an agent that can use sub-goal specifications to deal with sparse rewards. Oh et al. (2017) use sub-task instructions and hierarchical reinforcement learning to complete tasks with long action sequences.

In all these works, the text conveys the goal to the agent (e.g. 'move forward five steps'), thereby encouraging a direct connection between the instruction and the control policy. This tight coupling means that any grounding learned by the agent is likely to be tailored to the types of tasks seen in training, making generalization to a new distribution of dynamics or tasks challenging. In extreme cases, the agent may even function without acquiring an appropriate grounding between language and observations (Hu et al., 2019). In our setup, we assume that the text only provides high-level guidance without directly describing the correct actions for every game state.

**Language grounding by reading manuals**   A different line of work has explored the use of language as an auxiliary source of knowledge through text manuals. These manuals provide useful descriptions of the entities in the world and their dynamics (e.g. how they move or interact with other entities) that are optional for the agent to make use of and do not directly reveal the actions it has to take. Branavan et al. (2012) developed an agent to play the game of Civilization more effectively by reading the game manual. They make use of dependency parses and predicate labeling to construct feature-based representations of the text, which are then used to construct the action-value function used by the agent. Our method does not require such feature construction.

Narasimhan et al. (2018) and Zhong et al. (2020) used text descriptions of game dynamics to learn policies that generalize to new environments, without requiring feature engineering. However, these works assume some form of initial grounding provided to the agent (e.g. a mapping between object IDs and their descriptions, or the use of entity names in text as state observations). In contrast, our model figures out even this fundamental mapping between entity IDs in observation space and their symbols in text entirely through interaction with the environment.

## 3 FRAMEWORK

Our objective is to demonstrate grounding of environment dynamics and entities in a multi-task setup in order to drive generalization to unseen environments. Here and throughout, we refer to an *entity* as an object represented as a symbol in the observation that the agent may interact with.

**Environment** We model decision making in each environment as a Partially-Observable Markov Decision Process (POMDP) with the 8-tuple $(S, A, O, P, R, E, Z, M)$. $S$ and $O$ are the set of all states and observations respectively where each $o \in O$ contains entities from the set of entities $E$. At each step $t$, the agent takes some action $a_t \in A$. $P(s_{t+1}|s_t, a_t)$ is the transition distribution over all possible next states $s_{t+1}$ conditioned on the current state $s_t$ and action $a_t$. $R(s_t, a_t, s_{t+1})$ is a function that provides the agent with a reward $r_t \in \mathbb{R}$ for action $a_t$ and transition from $s_t$ to $s_{t+1}$. $Z$ is a set of text descriptions, with each $z \in Z$ providing information about an entity $e \in E$. $M$ is the map $z_e \mapsto e$ which identifies the entity that each description describes. $M$, $P$, and $R$ are not available to the agent. Note that there might not be a one-to-one mapping between $Z$ and entities in the current state observation.

**Reinforcement Learning (RL)** The objective of the agent is to find a policy $\pi : O \rightarrow A$ to maximize its cumulative reward in an episode. If $\pi$ is parameterized by $\theta$, standard deep RL approaches optimize $\theta$ to maximize the expected reward of following $\pi_\theta$. In our setup, we want the agent to learn a policy $\pi_\theta(a|o, Z)$ that conditions its behavior on the provided text. However, in contrast to previous work (Narasimhan et al., 2018; Zhong et al., 2020), $M$ is not available to our agent and must be learned through interaction.

## 4 MODEL

To learn a latent mapping between text symbols and entities, we develop a new model, EMMA (Entity Mapper with Multi-modal Attention), which employs a soft-attention mechanism over the text descriptions. At a high level, for each entity description, EMMA first generates key and value vectors from their respective token embeddings obtained using a pretrained language model. Each entity attends to the descriptors via a symbol embedding that acts as the attention query. Then, instead of representing each entity with its embedding, we use the resulting attention-scaled values as a proxy for the entity. This approach helps our model learn a control policy that focuses on entity roles (e.g. *enemy, goal*) while using the entities' identity (e.g. *queen, mage*) to selectively read the text. We describe each component of EMMA below and in Figure 2.

**Text encoder** Our input consists of a $h \times w$ grid observation $o \in O$ with a set of entity descriptions $Z$. We encode each description $z \in Z$ using a BERT-base model whose parameters are fixed throughout training (Devlin et al., 2018; Wolf et al., 2019). For a description $z$, let $t_1, ..., t_n$ be its token embeddings generated by our encoder. We obtain key and value vectors $k_z, v_z$:

$$k_z = \sum_{i=1}^{n} \alpha_i W_k t_i + b_k, \qquad \alpha = \text{softmax}\big((u_k \cdot t_j)_{j=1}^n\big) \tag{1}$$

$$v_z = \sum_{i=1}^{n} \beta_i W_v t_i + b_v, \qquad \beta = \text{softmax}\big((u_v \cdot t_j)_{j=1}^n\big) \tag{2}$$

The key and value vectors are simply linear combinations of $W_k t_i + b_k$ and $W_v t_i + b_v$ with weights $\alpha, \beta$ respectively, where $W_k, W_v$ are matrices which transform each token to $d$ dimensions and $b_k, b_v$ are biases. The weights $\alpha, \beta$ are obtained by taking the softmax over the dot products $(u_k \cdot t_j)_{j=1}^n$

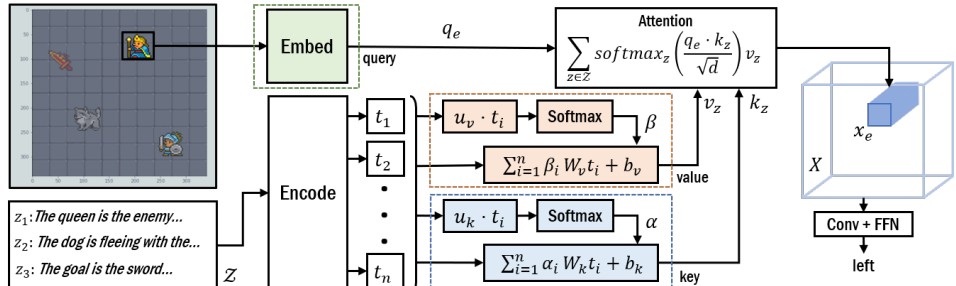

Figure 2: Schematic of our model EMMA, which creates a representation for entities using multi-modal attention over the observations and text manual. Mechanisms for the key, query, and value are shaded in blue, green, and red respectively.

and $(u_v \cdot t_j)_{j=1}^n$ respectively. These weights imbue our model with the ability to focus on relevant tokens. All of $W_k, b_k, u_k, W_v, b_v, u_v$ are learned parameters.

**Entity representation generator**  To get a representation for each non-avatar (non-player) entity $e$, we embed its symbol into a query vector $q_e$ of dimension $d$ to attend to the descriptions $z \in Z$ with their respective key and value vectors $k_z, v_z$. We use scaled dot-product attention (Vaswani et al., 2017) and denote the resulting representation for the entity $e$ as $x_e$:

$$x_e = \sum_{i=1}^m \gamma_i v_{z_i} \qquad \gamma = \text{softmax}\left(\left(\frac{q_e \cdot k_{z_j}}{\sqrt{d}}\right)_{j=1}^m\right) \qquad (3)$$

where $m = |Z|$ is the number of descriptions in the manual. This mechanism allows EMMA to accomplish two forms of language grounding: the key and query select relevant descriptions for each object by matching entities to names (e.g. *mage*), and the value extracts information relevant to the entities' behaviors in the world (e.g. *enemy, chasing*).

For each entity $e$ in the observation, we place its representation $x_e$ into a tensor $X \in \mathbb{R}^{h \times w \times d}$ at the same coordinates as the entity position in the observation $o$ to maintain full spatial information. The representation for the avatar (player entity) is simply a learned embedding of dimension $d$.

**Action Module**  To provide temporal information, we concatenate the outputs of the representation generator from the three most recent states to obtain a tensor $X' \in \mathbb{R}^{h \times w \times 3d}$. To get a distribution over the next actions $\pi(a|o, Z)$, we run a 2D convolution on $X'$ over the $h, w$ dimensions. The flattened feature maps are passed through a fully-connected feed-forward network terminating in a softmax over the possible actions.

$$y = \text{Flatten}\big(\text{Conv2D}(X')\big) \quad \pi(a|o, Z) = \text{softmax}\big(\text{FFN}(y)\big) \qquad (4)$$

In contrast to previous approaches that use global observation features to read the manual (Zhong et al., 2020), we build a text-conditioned representation for each entity $(x_e)$. One advantage is that $x_e$ can directly replace the entity embeddings typically used to embed the state observation in most models. This means EMMA's action module can easily be swapped with existing models, such as txt2pi (Zhong et al., 2020) while still being completely end-to-end differentiable. Further details about EMMA and its design can be found in Appendix D.

## 5  EXPERIMENTAL SETUP

### 5.1  TASK

**Motivation and Design**  We require a domain in which grounding the text descriptions $Z$ to dynamics and learning the mapping $M$ for all the entities in $E$ is necessary to obtain a good reward. Moreover, there must be enough environments to induce the mapping $M$.

With these requirements in mind, we devise a new multi-task domain *Messenger* using the Py-VGDL framework (Schaul, 2013). In *Messenger*, each entity can take on one of three roles: a *dangerous enemy*, a *secret message*, and a *crucial goal*. The player's objective is to bring the message to the goal while avoiding the enemy. If the player encounters the enemy at any point in the game, or the goal without first obtaining the message, it loses the game and obtains a reward of $-1$. Rewards of $0.5$ and $1$ are provided for obtaining and delivering the message to the goal respectively. Each of the enemy, message, and goal roles may be filled by one of twelve different entities, and each is assigned a *stationary, chasing*, or *fleeing* movement type to provide varying dynamics. Each set of entity-role assignments (henceforth referred to as a game) is initialized on a $10 \times 10$ grid. The agent can navigate via *up, down, left, right, and stay* actions and interacts with another entity when both occupy the same cell. Some game examples are presented in Figure 1.

The same set of entities with the same movements may be assigned different roles. Thus, two environments may have identical observations but differ in the reward function $R$ (which is not available to the agent) and the text manual $Z$ (which is available). Thus, our agent must learn to extract information from $Z$ to succeed consistently.

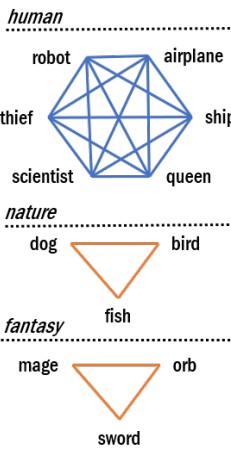

**Grounding Entities via Interaction**    In previous work (Zhong et al., 2020), every combination of entities is possibly observed during training. Thus, for an entity $e$, its symbol in the observation (e.g. ♕) is the only one that always appears together with its text symbol (e.g. *queen*). This co-occurrence provides a strong inherent bias towards the correct grounding without needing to act in the environment. We denote these games in which each entity can appear with every other entity as **multi-combination (MC)** games.

The MC assumption may not always be realistic in practice — some entities are very unlikely to appear together (e.g. *airplane, thief, sword*) while others may co-occur exclusively with each other (e.g. *mage, orb, sword*). We denote games in which the same three entities always appear together as **single-combination (SC)** games.

For SC games, any text symbol in the manual (e.g. *mage, enemy, the*, etc.) co-occurs the same number of times with all entities present, providing the agent no information about $M$. Thus, *the agent must ground*

Figure 3: Entities and their subdivision into *human, nature* and *fantasy* sub-worlds. Each $K_3$ subgraph is a combination of entities that may appear during training.

*these entities entirely via interaction*. That is, the agent must align the behavior of the entities with its grounding of *enemy, message, goal*. For example, if the agent has the message, interacts with entity $e$ and obtains a reward of $1$, it must infer from the description "*The mage is the goal*" that $e$ must be a *mage* (assuming it has grounded *goal* correctly).

We divide the entities in *Messenger* into *human, nature* and *fantasy* sub-worlds (Fig. 3) and exclude from training any games in which entities from different sub-world appear together. In particular, the *nature* and *fantasy* subworlds form SC games and the *human* subworld forms the MC games.

**Text Descriptions**    Unlike previous work on language grounding in grid environments (Zhong et al., 2020; Chevalier-Boisvert et al., 2019), we do not use templated or rule-generated text. We collected 3,881 unique free-form entity descriptions in English via Amazon Mechanical Turk (Buhrmester et al., 2016) by asking workers to paraphrase prompt sentences.

To increase the diversity of responses, the prompts were themselves produced from 82 crowdsourced templates When constructing the prompts, we intentionally inject multiple synonyms for each entity. Workers often further paraphrased these synonyms, resulting in multiple ways to describe the same entity (e.g. *airplane, jet, flying machine, aircraft, airliner* etc.). Learning to map these different text symbols to the same entity is another unique challenge in *Messenger*.

- the flying machine remains still, and is also the note of upmost secrecy.
- the airplane is coming in your direction. that airplane is the *pivitol* target.
- the winged creature escaping from you is the vital target.
- the fleeing *plan* is a critical target.

Table 1: Example descriptions for *Messenger*. Note the use of synonyms, multiple sentences per description, typos (*plane, plan*) and the need to disambiguate similar words (*flying machine, winged creature*).

Each training manual consists of a set of three descriptions, one for each enemy, message, and goal entities. In total there are over $1.3 \times 10^9$ possible manuals each with an average length of 30 words. The total vocabulary size of the free-form descriptions is 1,016. Besides lower-casing the worker responses, we do not do any preprocessing. Example descriptions can be found in Table 1. Further details regarding data collection can be found in appendix A.

**Train-Evaluation Split** Each entity is trained on two out of three possible roles, with the third role reserved for validation and testing. This forces models to make compositional entity-role generalizations for every entity to succeed on the evaluation games. In total we have 44 training, 32 validation, and 32 test games. We train on 2,557 of the text descriptions and reserve 652 each for validation and testing respectively.

**Comparison with Previous Environments** The primary focus of our work is on robust grounding of entities with rich language, a variety of entity synonyms, in an interactive setting without any priors or co-occurrence biases to help induce this grounding. To this end, *Messenger* features numerous challenges not found in prior work.

In Narasimhan et al. (2018), an oracle is used to concatenate the text representation to its corresponding entity representation. Access to such an oracle is a strong assumption in the wild and eliminates the need to ground the entities altogether.

In RTFM (Zhong et al., 2020), the observation is a grid of text which shares a set of symbols with the manual and both manual and observations are embedded into the same space. A key challenge unique to *Messenger* is learning to map between the observation and manual when they are embedded into different spaces. Furthermore, in RTFM all combinations of entities appear during training, which as we explained earlier, provides an additional signal that may simplify the grounding problem.

RTFM uses a small number rule-based templates to construct each manual, and each entity is referred to in a single way (e.g. *goblin* is always *goblin*). In contrast, *Messenger* features thousands of completely free-form descriptions and each entity may be referenced in multiple ways. For a more detailed comparison of RTFM and *Messenger*, including why we do not simply extend RTFM can be found in Appendix B.1.

## 5.2 BASELINES

**1) Mean-Bag of Sentences (Mean-BOS)** This is a variant of EMMA (our model) with the attention mechanism ablated. We average the value vectors obtained from equation 2 for each descriptor to obtain $\bar{v}$ which is used by the action module.

$$\bar{v} = \frac{1}{|Z|} \sum_{z \in Z} v_z \quad y = \text{Flatten}\big(\text{Conv2D}(\text{Emb}(o))\big) \quad \pi(a|o, Z) = \text{softmax}\big(\text{FFN}([y; \bar{v}])\big) \quad (5)$$

**2) Game ID-Conditioned (G-ID)** To assess the importance of language in our setup, we test a model with no language understanding on *Messenger*. We provide an auxiliary vector $I$ of integer IDs that reveals the mapping between entity symbols and roles (Fig. 4). These integers are embedded and concatenated to form the vector $v_I$ which is used by the action module to generate a distribution over the next actions:

$$y = \text{Flatten}\big(\text{Conv2D}(\text{Emb}(o))\big)$$
$$\pi(a|o, Z) = \text{softmax}\big(\text{FFN}([y; v_I])\big) \quad (6)$$

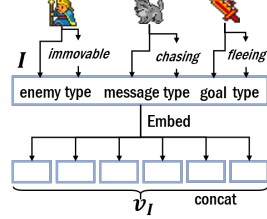

Figure 4: G-ID model

**3) Bayesian Attention Module (BAM)** This baseline uses a hard-attention mechanism with a naive Bayes classifier trained to learn $M$. This approach is similar to a word alignment model used in machine translation approaches such as the IBM Model 1 (Brown et al., 1993). Specifically, for some set of observed entities $E' \subseteq E$ in the current environment:

$$\text{BAM}(z, E') = \arg\max_{e \in E'} P(e|z) \qquad P(z|e) = \prod_{t \in z} P(t|e) \qquad P(t|e) = \frac{C(t, e)}{\sum_{t'} C(t', e)} \quad (7)$$

where $t \in z$ are tokens in $z$, $t'$ is any token in the manual vocabulary and $C$ refers to co-occurence counts. We use Bayes' rule to flip the conditional. We let $x_e = v_z$ from equation 2 for the $z$ that maps to $e$. If two descriptions map to the same entity, we take the one with higher $P(e|z)$, and if an entity receives no assignment we represent it with a learned default embedding $\text{Emb}(e)$. We pretrain BAM on $1.5 \times 10^6$ episodes.

**4) Oracle-Map (O-Map)**    To get a sense of the upper-bound on EMMA's performance, we consider a model that has access to the descriptor to entity map $M$, similar to Narasimhan et al. (2018). Thus, it is identical to EMMA except that the representation for each entity $x_e$ is obtained as in equation 8.

$$x_e = \sum_{z \in Z} \mathbb{1}[M(z) = e]v_z \tag{8}$$

### 5.3 TRAINING AND EVALUATION

**Curriculum**    Learning the entity groundings directly on *Messenger* is too difficult for the models we consider. Our initial experiments revealed that EMMA could not learn an entity grounding efficiently in this setting (Figure 5 (middle)). Thus, we introduce a two-stage curriculum to train our models (Bengio et al., 2009). In **stage 1 (S1)**, all entities begin two steps from the agent and are immovable. The agent either begins with or without the message and must interact with the correct entity. It is provided a reward of $1$ if it does so, and $-1$ otherwise. Models are first pretrained on S1, and all model parameters are transferred to **stage 2 (S2)**, where entities are mobile and the agent always begins without the message. In each training game there is one chasing, one fleeing and one immovable entity. On both S1 and S2 we train our models in a multi-task fashion by sampling a random game and appropriate manual at the start of each episode.

All models are end-to-end differentiable and we train them using proximal policy optimization (PPO) (Schulman et al., 2017) and the Adam optimizer (Kingma & Ba, 2014) with a constant learning rate of $5 \times 10^{-5}$. Additional details can be found in Appendix C.

**Evaluation**    On S1 test games, the entities start in the same locations as the training games and are also immovable. Thus, a model can apply the same policy used during training provided it can infer the (unseen) entity-role assignments, effectively testing the state estimation capabilities of models.

*On S2 test games, we introduce new combinations of object movements to test adaptation to new dynamics.* Specifically, compared to training there are two chasing entities during testing instead of one. We also consider a simpler state estimation test (S2-SE) where entity movements are identical to those seen during training. Environment details can be found in Appendix B.

## 6 RESULTS

### 6.1 MULTI-TASK PERFORMANCE

Figures 5 (left and middle) show rewards for both train and validation games as a function of training steps. The advantage of textual understanding is clear; on both S1 and S2, EMMA and the O-Map baseline converge to good policies much faster than the other

|  | G-ID | Mean-BOS | BAM | EMMA | O-Map |
|---|---|---|---|---|---|
| S1-All | $89 \pm 3.8$ | $\mathbf{90 \pm 7.2}$ | $84 \pm 1.3$ | $88 \pm 2.3$ | $97 \pm 0.8$ |
| S1-MC | $90 \pm 5.5$ | $91 \pm 6.5$ | $\mathbf{97 \pm 0.9}$ | $88 \pm 2.4$ | $97 \pm 0.3$ |
| S1-SC | $89 \pm 3.7$ | $\mathbf{90 \pm 6.8}$ | $51 \pm 1.6$ | $87 \pm 1.6$ | $96 \pm 0.6$ |
| S2-All | $3.6 \pm 0.6$ | $2.1 \pm 0.5$ | $69 \pm 1.1$ | $\mathbf{95 \pm 0.4}$ | $96 \pm 0.8$ |
| S2-MC | $3.4 \pm 0.7$ | $2.9 \pm 1.4$ | $85 \pm 0.9$ | $\mathbf{96 \pm 0.2}$ | $96 \pm 0.4$ |
| S2-SC | $3.9 \pm 1.5$ | $2.4 \pm 0.6$ | $22 \pm 4.8$ | $\mathbf{95 \pm 0.5}$ | $94 \pm 0.4$ |

Table 2: Win rates on training games. *All* denotes overall win rates, *MC* and *SC* denote multi and single-combination respectively.

baselines. However, EMMA trained directly on S2 with pretraining on S1 (EMMA-(no curriculum), Fig. 5) indicate that fitting directly onto S2 remains a challenge.

Table 2 details win rates on the training games, with a breakdown over single (SC) and multi combination (MC) games. We observe that the naive Bayes classifier can assign over $99\%$ of training descriptors correctly on MC games. However, on SC games that require interactive entity grounding, win rates are up to $60\%$ lower. Our model (EMMA) can consistently win on both multi and single combination games, demonstrating EMMA's ability to ground entities using interaction and reward signals alone, without co-occurrences statistics between entity and text symbols to guide its grounding.

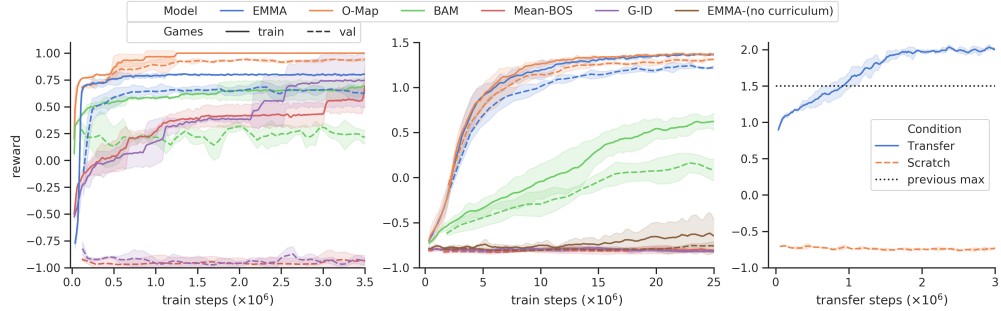

Figure 5: Average episodic rewards on S1 **(left)** and S2 **(middle)** on training *(solid line)* and validation *(dotted line)* games, as a function of training steps (x-axis). Reward is a combination of both single and multi-combination games. EMMA-(no curriculum) denotes EMMA trained directly on S2. We evaluate transfer performance of EMMA on games with novel entities and reward mechanics not found in *Messenger* **(right)**. Our model trained on *Messenger* (transfer) learns the new games much faster than a model trained from scratch (scratch). All results are averaged over three seeds and shaded area indicates standard deviation.

## 6.2 GENERALIZATION

|  | G-ID | Mean-BOS | BAM | EMMA | O-Map |
|---|---|---|---|---|---|
| S1 | $18 \pm 8.2$ | $6.7 \pm 2.8$ | $66 \pm 1.5$ | $\mathbf{85 \pm 1.4}$ | $97 \pm 0.3$ |
| S2 | $15 \pm 1.1$ | $16 \pm 1.4$ | $39 \pm 3.8$ | $\mathbf{79 \pm 1.9}$ | $82 \pm 1.7$ |
| S2-SE | $3.5 \pm 0.1$ | $1.7 \pm 0.7$ | $47 \pm 3.8$ | $\mathbf{90 \pm 1.8}$ | $93 \pm 0.4$ |

Table 3: Win rates on test games over three seeds. S1, S2 denotes stage one and stage two respectively. *SE* denotes state-estimation. EMMA can generalize to unseen games and almost matches the O-Map model on S2.

**New Entities** To assess EMMA's ability at picking up novel game mechanics which are not specified in the provided text, we introduce two new stationary collectibles into *Messenger* — a trap and gold which provide additional rewards of $-1$ and $1$ respectively. We finetune EMMA on the 32 validation games with these new entities. The model learns to collect the gold and avoid the trap while accomplishing the original objectives in *Messenger* (Figure 5 (right)). Compared to training from scratch, EMMA pretrained on the original *Messenger* games is able to achieve a much higher reward in this modified setting in the same amount of steps, exceeding the previous maximum reward of 1.5 in $1 \times 10^6$ steps.

**Test games** Results on test games are presented in Table 3. Both the G-ID and Mean-BOS baselines fail to generalize in all cases. The G-ID baseline has complete access to distinguishing information between games that is necessary to succeed. However it overfits to the entity-role assignments it has seen during training, resulting in poor test performance. BAM demonstrates some ability to generalize to test games, but performance on games with single-combination entities are considerably lower, bringing the average down.

|  | BAM | EMMA |
|---|---|---|
| Append | $34 \pm 0.2$ | $\mathbf{75 \pm 1.7}$ |
| Delete | $20 \pm 1.1$ | $\mathbf{36 \pm 1.8}$ |
| Synonyms | $11 \pm 0.5$ | $\mathbf{72 \pm 3.1}$ |

Table 4: Win rates on S2 test games over three seeds for *Append, Delete* and *Synonym* cases.

In contrast, EMMA can win $85\%$ and $79\%$ of test games on S1 and S2 respectively, almost matching the performance of the O-Map model. It also performs the best on the S2-SE test games. By extracting information from the relevant descriptor for each entity, EMMA is able to drastically simplify each task — it simply needs to learn a policy for how to interact with *enemy, messenger and goal* archetypes instead of memorizing a policy for each entity. This facilitates knowledge sharing between games, and generalization to unseen games.

## 6.3 ROBUSTNESS

**Test-Time** We first assess the robustness of trained BAM and EMMA models against text manual variations on S2 test games in table 4. We test each model's ability to: (1) handle an extra descriptor for an entity not found in the game (Append), (2) reason about the role of objects without a descriptor by deleting a sentence from the input at random (Delete) and (3) generalize to unseen synonyms (Synonyms). For the last case, we use (unseen) templated descriptions filled in with entity synonyms not seen during training.

Both models can retain their performance when presented with an extraneous description, and suffer considerably when a description is deleted. A key difference, however, is in our model's ability to generalize to unseen entity synonyms. EMMA wins almost 72% of games compared to 11% by the BAM model in this setting.

**Train-Time**  We test the ability to learn entity groundings with added neutral entities and negated descriptions (Table 5).

*Neutral entities.* At the start of each episode, we randomly select one of five neutral entities and insert it into the observation. The neutral entities are not described by the text, do not interact with the agent and provide no reward signal. The neutral entities are distinct from the objects in figure 3.

|         | Train        | Test        |
|---------|--------------|-------------|
| S1-Neu  | $92 \pm 1.0$ | $88 \pm 0.7$ |
| S2-Neu  | $95 \pm 0.4$ | $75 \pm 4.0$ |
| S1-Neg  | $87 \pm 3.8$ | $67 \pm 29$ |
| S2-Neg  | $88 \pm 8.8$ | $58 \pm 28$ |

Table 5: Percent win rates on train and test games for EMMA on the negation (Neg) and neutral (Neu) training cases on stage one (S1) and stage two (S2).

*Negation.* On each training episode with probability $0.25$ we select one description, negate it, and change the role. (e.g. *the mage is an enemy* becomes *the mage is not the message*). This case forces the model to consider the roles of the other two entities to deduce the role of the entity with the negated description. While EMMA can ground entities and performs well with neutral entities, it sometimes fails to ground the entities correctly with negated descriptions, affecting its performance on test games.

## 6.4 ANALYSIS OF GROUNDING

We visualize the attention weights for EMMA in Figure 6. Our model is always trained on a set of three descriptors, but to assess the overall latent mapping learned by our model, we evaluate the attention weights over 12 descriptions, one for every entity. EMMA is able to place most weight for entity $e$ onto its descriptor $z_e$. In particular, EMMA learns a grounding for *dog, bird, fish, mage, sword and orb* — entities for which co-occurrence statistics provide no meaningful alignment information, demonstrating that our model can learn groundings for these entities via interaction alone. This also hints that the acquired grounding can enable EMMA to comfortably scale up to environments containing larger sets of entities.

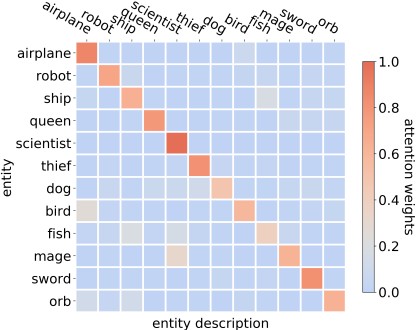

Figure 6: Attention weights for EMMA computed from equation 3. Each row shows the attention weights for one entity over 12 randomly selected descriptors, each of which describe a separate entity indicated by the column label.

## 7 CONCLUSION

In this paper, we develop a new model, EMMA (Entity Mapper with Multi-modal Attention) to leverage textual descriptions for generalization of control policies to new environments. Unlike prior work, we do not assume access to any form of prior knowledge connecting text and state observations, and learn both symbol grounding and control policy simultaneously. EMMA employs a multi-modal entity-conditioned attention module and learns a latent grounding of entities and dynamics using only environment rewards. Our empirical results on a newly developed multi-task game framework with crowdsourced text manuals demonstrate that EMMA shows strong generalization performance and robust grounding of entities. We hope that these results can lead to further research in enabling generalization for RL using natural language.

**Applications and Future Work**  Studying entity grounding in interactive settings (with humans or other agents) will be important for agents that act in the world and automatically learn new concepts. While EMMA generalizes well to unseen games, its performance when additional reasoning is required (negation, deleted descriptions) is lower and warrants further study. Furthermore, training EMMA directly on *Messenger* without a curriculum results in very poor performance. This demonstrates that learning stable entity groundings on *Messenger* remains a difficult problem with long trajectories and sparse rewards (training directly on S2) and when additional reasoning is required (negation). We leave this to future work.

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

## A   TEXT MANUAL

| **Example Input** |
|---|
| - The bird that is coming near you is the dangerous enemy. |
| - The secret message is in the thief's hand as he evades you. |
| - The immovable object is the mage who holds a goal that is crucial. |

| **Enemy Descriptions** |
|---|
| Adjectives: dangerous, deadly, lethal |
| Role: enemy, opponent, adversary |

| **Message Descriptions** |
|---|
| Adjectives: restricted, classified, secret |
| Role: message, memo, report |

| **Goal Descriptions** |
|---|
| Adjectives: crucial, vital, essential |
| Role: goal, target, aim |

Table 6: Example template descriptions. Each underlined word in the example input indicate blanks that may be swapped in the template. Each template takes a word for the object being described (bird, thief, mage), its role (enemy, message, goal) and an adjective (dangerous, secret, crucial).

To collect the text manual, we first crowdsource 82 templates (with 2,214 possible descriptions after filling in the blanks). Each Amazon Mechanical Turk worker is asked to paraphrase a prompt sentence while preserving words in boldface (which become the blanks in our templates). We have three blanks per template, one each for the entity, role and an adjective. For each role (enemy, message, goal) we have three role words and three adjectives that are synonymous (Table 6). Each entity is also described in three synonymous ways. Thus, every entity-role assignment can be described in 27 different ways on the same template. Raw templates are filtered for duplicates, converted to lowercase, and corrected for typos to prevent confusion on downstream collection tasks.

To collect the free form text for a specific entity-role assignment, we first sample a random template and fill each blank with one of the three possible synonyms. The filled template becomes the prompt that is shown to the worker. Aside from lower-casing the free-form descriptions, we do no further pre-processing.

On all tasks (template and free-form) we provide an example prompt (which is distinct from the one provided) and examples to provide additional task clarity. For each prompt, we obtain two distinct paraphrased sentences to promote response diversity. To ensure fluency in all responses, we limited workers to those located in the United States with at least 10,000 completed HITs and an acceptance rate of $\geq 99\%$. On all data collection tasks, we limit a single worker to a maximum of 36 responses to ensure worker diversity. In total, 297 workers participated in the data collection process.

## B   ENVIRONMENT DETAILS

Details about *Messenger* can be found in table 7. On stage 1 (S1), the three entities start randomly in three out of four possible locations, two cells away from the avatar. The agent always begins in the center of the grid. It starts without the message with probability $0.8$ and begin with the message otherwise. When the avatar obtains the message, we capture this information by changing the avatar symbol in the observation. On S1, we limit each episode to four steps and provide a reward of $-1$ if the agent does not complete the objective within this limit.

| Objects | bird, dog, fish, scientist, queen, thief, airplane, robot, ship, mage, sword, orb |
|---|---|
| Roles | enemy, message, goal |
| Movements | chasing, fleeing, immovable |
| Total games | $P(12, 3) = 1320$ |
| Total variants | $1320 \times 3! = 7920$ |
| Initial States/variant | 24 |

Table 7: Basic information about our domain *Messenger*. Each game features 3 out of 12 possible non-player entities, each assigned a role of enemy, message or goal. Each training game has 3! variants corresponding to the assignment of chaser, fleeing and immovable movement types to each entity.

On stage 2 (S2), the avatar and entities are shuffled between four possible starting locations at the start of each episode. On S2, the mobile entities (fleeing, chasing) move at half the speed of the agent. On S2, we limit each episode to 64 steps and like in S1, we provide a reward of $-1$ if the agent does not complete the objective within this limit.

Since there are only 4 single-combination (SC) training games and 40 multi-combination (MC) training games, we sample the games non-uniformly at the start of each episode to ensure that there is enough interaction with SC entities to induce an entity grounding. On both S1 and S2, we sample an SC game with probability $0.25$ and an MC game otherwise. Not all descriptions have movement type information (e.g. "chasing"). We also collect *unknown type* descriptions with no movement type information. During training, each description is independently an unknown type description with probability $0.15$

**Negation**  We procedurally generate the negated text by negating existential words (e.g. "is an enemy" becomes "is not an enemy"). We manually negate those descriptions not captured by the rules. During both training and evaluation, we provide a complete text manual without any negated description with $0.75$ probability, and randomly select a description in the manual to negate otherwise. When we negate an entity description $z_e$ to $z'_e$, we also change the role ("...is an enemy" becomes "...is not a goal", for example). Thus the information present in the manual has not changed, but the agent must look at the remaining two descriptions to deduce the role of $e$ with description $z'_e$.

**Transfer Learning**  We test transfer by introducing two new entities – a trap and a gold which provide rewards of $-1$ and $1$ respectively. Both collectables are randomly shuffled between two possible starting locations at the start of each episode and do not move. We train the models in this new setting in a multi-task fashion on the 32 validation games. After the agent encounters either the trap or gold, the collected item disappears. Neither item terminates the episode and the agent can still win or lose the current episode regardless of whether it has picked up the gold or trap.

### B.1 COMPARISON WITH RTFM

The main novelty of our work (both the *Messenger* environment and our model) is in specifically tackling the issue of *entity grounding* without any prior knowledge. To do this, *Messenger* in contrast to RTFM (1) does not have any signal connecting entities to text symbols, (2) features much richer language, and (3) requires interaction in the environment to ground entities to text. We describe these in more detail:

1. RTFM's observation space consists of a grid of text which shares a set of symbols with the text manual. Thus, both the text in the manual and the observation are embedded into the same space (e.g. using the same word vectors), essentially providing models with the entity grounding upfront. In contrast, our environment has a separate set of symbols for the entities with no relation to the text in our manual. Thus, the entities and text are embedded into different spaces, and learning to map between these two spaces is the key challenge in our environment that has not been explored before.

2. RTFM features only 32 total rule-based templates for the text, and each entity can only be referred to in a single way (*goblin* is always *goblin*). In contrast, we crowdsourced thousands of completely free-form descriptions in *two* rounds using Amazon Mechanical

Turk. After obtaining the seed templates from the first round, we intentionally inject multiple synonyms for each entity to construct each prompt for the second round. Workers often further paraphrased these synonyms, resulting in 5, 6 or often more ways to describe the same entity (e.g. *airplane, jet, flying machine, aircraft, airliner* etc.). The need to map these different text symbols to the same entity further complicates the entity grounding problem in our case and more closely mirrors the challenges of grounding entities in the real world. We believe *Messenger* provides a much closer approximation to "natural" language compared to RTFM.

3. RTFM features all possible combinations of entities during training, which as we explained in Sec. 5.1 provides an additional signal that may simplify the grounding problem.

4. Furthermore, each entity in RTFM only moves in a single way, whereas in *Messenger*, each entity may have different dynamics such as fleeing, chasing, and immovable entities (and this is also described in the text). This also allows us to test our model's ability to generalize to unseen dynamics with unseen entity movement combinations, whereas in RTFM the evaluation on unseen games is essentially state-estimation.

*Messenger* is quite similar to RTFM and shares several aspects (e.g. grid-world with different entities and goals). That said, there are numerous reasons why we were not able to adapt the original RTFM environment to meet our requirements. We enumerate them here:

1. The dynamics in RTFM make entity grounding (the primary focus of our work) difficult. *Messenger* requires much simpler reasoning than RTFM, and it is already too difficult to ground entities directly in *Messenger* without a curriculum. RTFM sidesteps the issue by providing this grounding beforehand.

2. Obtaining enough crowdsourced descriptions is hard with RTFM because of the more complicated dynamics. In RTFM, there are monsters, weapons, elements, modifiers, teams, variable goals and different weaknesses between entity types that need to be specified. Collecting enough descriptions that are entirely human written would be challenging. (RTFM sidesteps this issue by using templates to generate their text manual). In contrast, there are only entities, 3 roles, and a fixed goal in *Messenger*, making the text-collection task much more tractable.

3. The entities in our *Messenger* environment are carefully chosen to make entity grounding harder. In RTFM, each entity is referred to in a single way, and it is not clear how to refer to them in multiple ways (e.g. there are not too many other ways to say *goblin*). In contrast, we specifically chose a set of entities that allowed for multiple ways of description, and actively encouraged this during data collection.

4. The combination of entities that appear during training in *Messenger* is carefully designed. This is so that we can introduce single-combination games and the associated grounding challenges that come with it.

5. We have different movement types for each entity. These different movements are referred to in our text manual and significantly increase the richness and variety of descriptions we collected, and also allow us to test generalization to unseen movement combinations. In RTFM, the entity movements are the same and fixed for all entities.

6. Each entity's attribute is referenced in the observation in RTFM, e.g. the grid has entries such as *fire goblin*. We could add to the cell an extra symbol for *fire*, but this further obfuscates the entity grounding problem we are focusing on, because we would also need to obtain a grounding for all the attributes such as *fire*.

## C  IMPLEMENTATION DETAILS

For all experiments we use $d = 256$. When multiple entities $E'$ overlap in the observation, we fill the overlapping cell with the average of the entity representations $\frac{1}{|E'|} \sum_{e \in E'} x_e$. The convolutional layer consists of $2 \times 2$ kernels with stride $1$ and $64$ feature maps. The FFN in the action module is fully-connected with 3 layers and width of $128$. To give the Mean-BOS and G-ID baselines the ability to handle the additional conditioning information, we introduce an additional layer of width

512 at the front of the FFN for those baselines only. Between each layer, we use leaky ReLU as the activation function. All models were trained for a maximum of $3 \times 10^6$ episodes on S1 and 36 hours on S2. All experiments were conducted on a single Nvidia RTX2080 GPU.

## D  MODEL DESIGN

The weights $u_k$ and $u_v$ were introduced to make sure that the token embeddings for filler words such as *the, and, or* do not drown out the words relevant to the task when we take the average in equations 1 and 2. Qualitatively, we observe that $u_k$ learns to focus on tokens informative for identifying the entity (e.g. *mage, sword*) while $u_v$ learns to focus on tokens that help identify the entities' roles (e.g. *enemy, message*).

We also found that using a pretrained language model was critical for success due to the large number of ways to refer to a single entity (e.g. *airplane, jet, flying machine, aircraft, airliner* etc.).

### D.1  MODEL VARIATIONS

We consider a variation to EMMA. Instead of obtaining token weights $\alpha, \beta$ in equations 1 and 2 by taking a softmax over the token-embedding and vector products $u_k \cdot t$ and $u_v \cdot t$, we consider independently scaling each token using a sigmoid function. Specifically, we will obtain key and value vectors $k_z$ and $v_z$ using:

$$k_z = \sum_{i=1}^{n} \frac{\sigma(u_k \cdot t_i)}{\sum_{i=1}^{n} \sigma(u_k \cdot t_i)} W_k t_i + b_k \tag{9}$$

$$v_z = \sum_{i=1}^{n} \frac{\sigma(u_v \cdot t_i)}{\sum_{i=1}^{n} \sigma(u_v \cdot t_i)} W_v t_i + b_v \tag{10}$$

where $\sigma$ is the sigmoid function, and all other details are identical to EMMA. We call this model EMMA-$\sigma$. We notice that both EMMA and EMMA-$\sigma$ are able to obtain good training and validation performance, whith EMMA-$\sigma$ obtaining higher rewards on S2. However, on S1, EMMA is able to obtain a higher validation reward faster (Fig. 7). Moreover, EMMA can learn robust groundings even with neutral entities, while EMMA-$\sigma$ often overfits to a spurious grounding with neutral entities (Fig. 8). Although the independent scaling in EMMA-$\sigma$ allows the model to consider more tokens simultaneously, the softmax selection of EMMA facilitates more focused selection of relevant tokens, and this may help prevent overfitting.

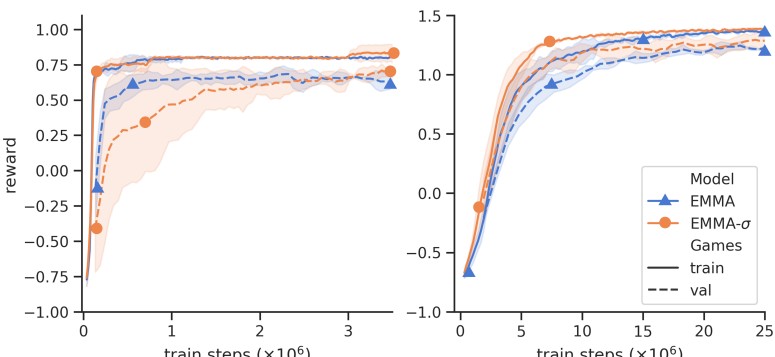

Figure 7: Average episodic rewards on S1 **(left)** and S2 **(right)** on training *(solid line)* and validation *(dotted line)* games, as a function of training steps (x-axis) for both EMMA and EMMA-$\sigma$. Both models are able to perform well, however, EMMA is able to obtain a good validation reward faster. All results are averaged over three seeds and shaded area indicates standard deviation.

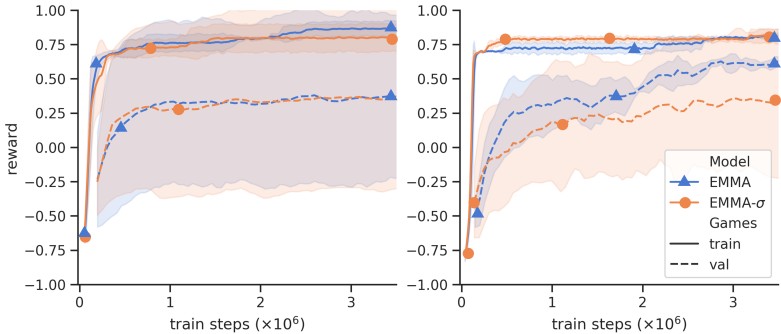

Figure 8: Average episodic rewards on S1 games with negation **(left)** and neutral entities **(right)** on training *(solid line)* and validation *(dotted line)* games, as a function of training steps (x-axis) for both EMMA and EMMA-$\sigma$. Both models struggle on negation, but EMMA is able to perform well with neutral entities. All results are averaged over three seeds and shaded area indicates standard deviation.

