# OpenReview forum: "Grounding Language to Entities for Generalization in Reinforcement Learning"
_ICLR.cc/2021/Conference — Reject_

### Official Review · AnonReviewer2 · 2020-10-27

**Rating:** 6
**Confidence:** 3

**Review:**

The paper presents a model for entity grounding from its textual description for a text-based language game. To test their new model named EMMA (Entity Mapper with Multi-modal Attention), they also present a new toy game framework and crowdsourced data composed of 1320 games (from 3,881 entity descriptions with crowdsourcing). In this game, the agent has to associate an entity with its description. The game is described in Figure 1 and section 5.1: there are three entities to each game (messenger, enemy, goal), and each entity can be stationary, chasing, or fleeing movement. The game has a one-sentence description for each entity (if I’m interpreting it correctly). The space of action is up, down, left, right, and stay. The goal of the player is always static — bring a message to the goal while not touching the enemy. From what I understand, the game here itself is substantially simpler than that of previous work (Zhong et al 20), which involves varying goals and modifiers.

Their architecture deviates from existing approaches (Zhong et al 20, Narasimhan et al 18) which did not require the model to learn a mapping from an entity to its description either by providing a mapping between objects and their textual descriptions or using entity names plainly. This is the main focus of the model — to learn a mapping between entities and their descriptions. The evaluation is also done in a manner that between train/validation/testing there’s no overlap between entity - role combination.

The model architecture is pretty standard, using a bert-base encoder with attention. Figure 2 clearly explains their model, where each character takes a weighted sum of descriptions as value vectors to decide the action.

The game feels a bit retrospectively designed to test the research idea. It would have been a lot more convincing if they use their model on the existing game environment, instead of creating a whole new game specifically designed to test their idea. The paper is clearly written and easy to follow, except for some parts (BAM parts weren’t easy to follow).

The paper presents nice studies throughout section 6, looking into learning in a more challenging evaluation set up such as generalization and supplementary material is also thorough.

For Figure 5, graphs are on the union of single and multi-combination games?
How much does it help to use pre-trained BERT encoder vs. training from scratch?

I’m not sure what future work can be done in this space. The model seems to be doing well in this constrained setting (Table 2). Could the authors provide how this model can be applied to more challenging and complex game scenarios, where there's no one-to-one mapping between entity description and an entity, and the goal is more complex?

---

> ### Author Response · Authors · 2020-11-13
> **Response**
>
> Thank you for your comments. We hope that our responses clarify some of the unique challenges we address that are not found in Zhong et al. (2020), as well as the opportunities to expand on our current work in the future.
>
> *There are three entities to each game (messenger, enemy, goal).* This may be a typo but just to be sure: there are 12 entities in total, each of which can be assigned one of the three roles you listed above.
>
> **The game has a one-sentence description for each entity (if I’m interpreting it correctly)**
>
> For the base environment, each entity has one description (which can be multi-sentence, see Table 1 for some examples). We also evaluate with additional and deleted descriptions (Table 4).
>
> **From what I understand, the game here itself is substantially simpler than that of previous work (Zhong et al 20), which involves varying goals and modifiers.**
>
> This is not entirely true. While RTFM (Zhong et al. 2020) has more complex game mechanics, our environment is focused on different challenges that reflect the difficulties of grounding entities in the wild not captured by RTFM. We highlighted some of these in the general comments and will also do so in our revision.
>
> **It would have been a lot more convincing if they use their model on the existing game environment.**
>
> For the reasons described in the general comments, we found modifying RTFM to be infeasible for answering our research questions, which are orthogonal to those RTFM is designed for. Future environments and approaches can merge the challenges found in these two environments.
>
> **For Figure 5, graphs are on the union of single and multi-combination games? How much does it help to use pre-trained BERT encoder vs. training from scratch?**
>
> You are right, in Fig. 5 the graphs are a combination of multi and single combination game performance. Table 2 has a more detailed breakdown of performance on single and multi combination games. It’s important to use a pre-trained BERT encoder because of the large number of synonyms in the descriptions for each entity. We found that training without language priors results in very poor performance.
>
> **I’m not sure what future work can be done in this space.**
>
> There are still a lot of open problems in Messenger. For one, no model that we tested can fit to the training games on the full Messenger environment without first training on a severely limited version (see the general comments on curriculum learning in Messenger). Performance with negated and deleted descriptions which require models to conduct additional levels of reasoning also severely reduce model performance.
>
> Furthermore, once we develop better techniques to fully solve Messenger, one can extend such environments to beyond grid worlds, e.g. 3D navigation, robotic control, etc. Another important challenge is the one you mention (below) —- learning from more generic descriptions without assuming 1-1 correspondence with entities.
>
> **Could the authors provide how this model can be applied to more challenging and complex game scenarios, where there's no one-to-one mapping between entity description and an entity, and the goal is more complex?**
>
> Our model should be able to handle multiple descriptions corresponding to the same entity (we expect the attention weight for the entity to be distributed evenly between its descriptions), although we have not tested this. For the same description corresponding to multiple entities, we could train a separate module to split these into multiple descriptions. Alternatively, we might try to use multi-head attention where the keys and values are obtained from the description, and the query is a learned entity embedding (which can learn to attend to the relevant tokens for that entity).
>
> One of the advantages of our approach is that we generate text-conditioned representations for each entity, which can directly replace the entity embeddings typically used to embed the state observation in most models. This means that the representations from EMMA can feed directly into existing models, such as txt2pi used in Zhong et al. (2020), which can be used to handle variable goals and multi-hop reasoning, while still being completely end-to-end differentiable.

---

### Official Review · AnonReviewer4 · 2020-10-28
**Solid extension of prior methods**

**Rating:** 7
**Confidence:** 3

**Review:**

## Summary
This is a significantly improved extension of prior work in this area. The authors significantly improve upon the previous techniques: see below. I have a few questions on clarity but overall the paper was well-presented.

## Quality & Clarity
The paper is generally fairly clear. There are a lot of details in this paper and some of them did not quite feel fully developed (understandable given the space constraints), such as justification for certain modeling choices.

For example, in section 4 under “Entity representation generator”, the authors say:
>  For each entity e in the observation we place its representation xe into a tensor X at the same coordinates as the entity position in the state s to maintain full spatial information
This implies to me that tensor X is a 2D grid matching the coordinate system. The next section says that the three most recent observations are concatenated together and a 2D convolution is run over them. Which of the three potential dimensions are used in the 2D convolution and which is flattened: the x/y coordinates or the time dimension?

In Section 4: Model, Text Encoder, you say that “These [alpha and beta] weights imbue our model with the ability to focus on relevant tokens”. Can you provide an intuition for what u_k and u_v are learning?

In Section 6.1: Multi-task Performance, I was not quite able to grasp why the single-combination tasks are more difficult. Can you explain the difference between these two kinds of tasks a bit more?

Overall though, I was able to easily understand the main points of the paper and the data was clearly visualised in graphs and tables.

## Originality & Significance
There are a lot of good contributions in this paper:
- natural language crowd-sourced manuals with two layers of “turk-indirection”: not only was the text crowdsourced but first the templates for the texts were crowdsourced
- a comparison of several different techniques with different attention mechanisms
- experiments with additional evaluations of generalisation and robustness, including more difficult test scenarios, adding neutral distractor entities, fine-tuning with additional sources of punishment/reward, negating the text, and replacing entities with synonyms
- a visualization of attention to demonstrate that the attention mechanism is working properly

These are original and significant contributions that go beyond what was done prior.

**Suggestion**: While I didn’t understand the difference in difficulty between the multi-combination / single-combination games, what I did hope to see when they were presented was an evaluation of how the agent was able to transfer learning from one of these enabled transfer to the other (e.g. learning in the fantasy setting transferred to the nature setting). This would demonstrate that the kind of structure present in the tasks is learned and can be applied to different kinds of entities / a different vocabulary.

---

> ### Author Response · Authors · 2020-11-13
> **Response**
>
> Thank you for your comments. We are happy you found that our work is a significantly improved extension of prior work in this area.
>
> **Clarification on the entity representation generator.**
>
> X is a tensor constructed from the current observation with dimension (h,w,c) where (h,w) are the height and width of the grid, and c is the embedding dimension (dim of xe). The tensors corresponding to the three most recent observations are concatenated along the c dimension, and the convolution is over the (h,w) dimensions with the c dimension as the channels and the resulting tensor is flattened to a single dimension.
>
> **Can you provide an intuition for what u_k and u_v are learning?**
>
> Qualitatively, we observe that u_k learns to put most weight onto tokens that are informative for identifying the entity (e.g. “mage”, “sword”) and u_v learns to put weight onto tokens that help identify the entities’ roles (e.g. “enemy”, “message”).
>
> **Why are the single-combination tasks more difficult?**
>
> We provided some clarification on the difference between single and multi combination tasks in the general comments and will do so in our revision as well.
>
> **Suggestion: Why not show transfer learning from one of these domains enabled transfer to the other (e.g. learning in the fantasy setting transferred to the nature setting).**
>
> To transfer to a different setting (e.g. nature) not observed during training would require the model to induce a mapping between text and entity symbols for entities it has not observed before (so the entity symbol embeddings would still be randomly initialized). This is a very interesting idea but introduces additional challenges, and future work can explore incorporating visual/text priors to induce the entity groundings in unseen domains.

---

### Official Review · AnonReviewer1 · 2020-10-29
**accurately executed language-conditioned RL paper, but somewhat too toy-ish**

**Rating:** 6
**Confidence:** 3

**Review:**

The paper considers the task of training an agent to act following a manual expressed in natural language. The manual describes the roles and the behaviors of the entities in the environment. Each entity can be the goal, the message or the enemy and it can also be either fleeing, chasing or not moving. The agent has to bring the message to the goal while avoiding the enemy. A model called EMMA is proposed to change entity representations based on the manual, thereby making the agent aware of the entities’ roles. It is shown that EMMA is more effective than simple baselines.

The paper is mostly clearly written.  The proposed model makes a lot of sense for the considered task. The paper can be seen as another proof-of-concept paper for acting based on a manual, but it should be noted that it is not the first one of its kind [1].

Some things were not entirely clear to me.

- In several places in the paper it is mentioned that the model learns “mapping between entity IDs in observation space and their symbols in text entirely through *interaction* with the environment.” I am not sure what interaction means here. Looking at the formulas, my understanding is that EMMA establishes a correspondence between the observed object symbols and the entities based on the entities’ descriptions. What role does interaction play in determining the object-role mapping? Same question about a similar sentence in Paragraph “Multi-Task Setup” of Section 5.1.
- Why is G-ID baseline performing so poorly on validation and test data? Doesn’t it have the complete access to the roles and behaviour of every object?
- Why is the paper not using a setup from the prior work, e.g. from [1]?

While the paper is clear and seems to be very accurately and correctly executed, I have concerns (which might be considered subjective) about the limited real-world impact of this kind of work. The paper does not discuss how this work is related to any real world RL applications, such as e.g. robotics or even building RL agents for games that people actually play. It does not discuss what the manual would be in these cases. Would it look anything like the description of the objects’s roles? It appears common in the field of RL to not ask for such justifications and happily accept gridworld studies motivated by generalization to “new environments”. But I would like to flag that for an outsider this disconnect from reality may seem troublesome.

---

> ### Author Response · Authors · 2020-11-13
> **Response**
>
> Thank you for your comments. We hope our answers will convince you that Messenger is a difficult environment, substantially different from RTFM, and a first step in a direction of research with real world impact.
>
> **My understanding is that EMMA establishes a correspondence between the observed object symbols and the entities based on the entities’ descriptions. What role does interaction play in determining the object-role mapping?**
>
> EMMA does establish a correspondence between the observed object symbols and the entities in text based on the descriptions. However, to learn this correspondence, EMMA must interact with the environment to obtain rewards, understand which types of objects provide rewards and thereby understand the grounding for them. We provided some clarification in the general comments (on single and multi combination games) and will make sure to clarify this in our revision.
>
> **Why is G-ID baseline performing so poorly on validation and test data? Doesn’t it have the complete access to the roles and behaviour of every object?**
>
> The G-ID baseline has complete access to all information necessary to succeed. However it overfits to the entity-role assignments it has seen during training. The training and evaluation split in Messenger is such that, if entity e is role r in evaluation, the agent never sees e as r during training. Thus, agents are prone to adopting the assumption that “e is never r”. Overcoming this pitfall requires models to completely decouple the entity from the role to make compositional generalizations, and the G-ID (and Mean-BOS) baselines fail to do this. This further demonstrates the difficulty of robust entity grounding in Messenger without overfitting. We will add some discussion on why G-ID fails to generalize in our revisions.
>
> **Why is the paper not using a setup from the prior work, e.g. from [1]?**
>
> Please see the general comments regarding why we do not use RTFM to evaluate entity grounding. The unique challenges of grounding entities made modifying RTFM to fit our needs infeasible.
>
> **The paper does not discuss how this work is related to any real world RL applications, such as e.g. robotics or even building RL agents for games that people actually play.**
>
> We believe studying entity grounding in interactive settings (with humans/other agents) will be important for agents that act in the world and automatically learn new concepts. For example, suppose an agent acting in the real world needs to ground “apple” and “orange”. If there are cases when we reference an “apple” without an “orange” present, the agent can disambiguate between the two fruits (analogous to how all entity combinations are available in RTFM). If however, the agent always sees the two fruits together, it needs additional information. We can say “oranges are squishier than apples”, but then the agent must interact with the entities, see which one aligns with its grounding of “squishy” to make the appropriate grounding of “orange”. (This setting is analogous to single-combination games in Messenger). Currently no other environment or approach we are aware of tests this form of grounding, and we hope Messenger represents a first step of many in this research direction.
>
> We will add in a discussion on this in our revision.

---

### Official Review · AnonReviewer3 · 2020-10-29
**An overall clear and straightforward paper based on an educated guess.**

**Rating:** 5
**Confidence:** 1

**Review:**


Natural language grounding is an interesting research direction and has attracted many researchers in recent years. Previous work mainly considered grounding the text to image objects. This paper considers collaboratively to learn “entity” representations and natural language explanations with a reinforcement learning framework. Specifically, a multi-modal attention network is proposed to model the interaction between the entity representation and the text descriptions. The entire framework is trained over multiple games in a multi-task manner. Experiments are conducted on a newly designed benchmark. The proposed RL framework achieves reasonable performance in domain games (training & test are from the same games) and also has a strong zero-shot generalization to unseen games and "entities" (thanks to the parameter sharing and multi-task learning). Besides, the newly released dataset may facilitate future research in natural language grounding.

 I am not familiar with the natural language grounding literature, so this might be an educated guess, which is listed as follows,


- The writing is relatively clear, although some specific illustrations are sometimes hard to follow. For example, the term “entity” is very ambiguous. In the NLP field, an entity might relate to a phrase that has specialized semantic meaning. I read several recent papers in the natural language grounding field but still puzzled about the term (for example, the semantic meaning of “entity” in  [2] is different from this paper).

- After reading this paper, I am not sure which part is the main novelty. The proposed reinforcement learning framework is relatively standard (no specialized designed state, policy, and reward). The interaction module is designed in a slightly straightforward way. Utilizing parameter sharing to encourage stronger generalization ability also seems reasonable. The novelty might be further clarified.
- The proposed approach seems to be similar to [1] (cited). The general idea is to learn a parameterized policy model that inputs the pair (entity [this paper] or visual representation [1] and text) and outputs the action of the agent in the game (correct me if I am wrong). Depending on different scenarios, the encoding network can be slightly different. It might be beneficial to have a detailed discussion of the difference between the methods proposed in the two papers.
- In Figure 5, the reward of different approaches seems to have small differences. However, the final winning rate of games varies a lot. It might be beneficial to have some discussions on this part.
- If my understanding is correct, the contribution of the paper is mainly focused on the strong generalization ability of the proposed approach across different games. However, it may need to clarify which part of the proposed approach contributes to the most significant influence.

Other minor points:
 - I am interested in the O-Map baseline, which seems to be a very good upper bound of the proposed approach, I am surprised to see that the EMMA approach outperforms the O-Map in the training settings. Some intuitive explanations might be useful. The second question is that, though utilizing the upper bound (O-Map), the winning rate still has some room to improve. It is suggested to have some further analysis.
 - I do not quite understand why the overall training process contains two rounds of stages. Is S1 stage training a warm-up training?

I would like to hear the authors’ responses to make my decision.

[1] Zhong et al. RTFM: Generalising to New Environment Dynamics via Reading. ICLR 2020

[2] Lai et al. Contextual Grounding of Natural Language Entities in Images. NeurIPS 2019 workshop

---

> ### Author Response · Authors · 2020-11-13
> **Response (1/2)**
>
> Thank you for your comments. We hope our responses will clarify some of the novelty in our work and help you make an informed decision.
>
> **Previous work mainly considered grounding the text to image objects.**
>
> In prior work such as Narasimhan et al. (2018) and Zhong et al. (2020), the observations are a grid of symbols, similar to what we use in Messenger. The difference is that in Zhong et al. (2020) the observation is actually a grid of text which shares symbols with the text manual (e.g. the symbol “goblin” is used in the grid, and the same symbol “goblin” is used in the manual). This tightly couples the state observation and the manual.
>
> **The writing is relatively clear, although some specific illustrations are sometimes hard to follow. For example, the term “entity” is very ambiguous.**
>
> We refer to an entity as a non-player object in the game that the agent can possibly interact with (represented as a symbol in the observation). We will clarify this in the paper. In addition, could you possibly point us to the illustrations you found hard to follow so that we can improve them? Thanks!
>
> **After reading this paper, I am not sure which part is the main novelty.**
>
> The main novelty is introducing an environment (Messenger) with a significantly harder entity-grounding problem compared to previous environments (such as [1]) and a model which solves Messenger (with pre-training on a simpler version). Please also see general comments - we further clarified some of the novelty and will make sure to highlight these in our revisions.
>
> **The proposed approach seems to be similar to [1] (cited). The general idea is to learn a parameterized policy model that inputs the pair (entity [this paper] or visual representation [1] and text) and outputs the action of the agent in the game.**
>
> In [1], the “visual representation” is actually a grid of text. The objectives of both approaches is to learn a text-conditioned policy. However, the challenges present in both environments are quite different and the approaches are designed to address the unique challenges in each. In Messenger, the game is simpler than [1], but there are no priors connecting the state observation to text, making entity grounding difficult. Our model handles this by learning to select relevant descriptions and build a text-conditioned representation for each entity whereas in [1], global observation features are used to read the text. Please also see general comments for a detailed discussion on this.
>
> **In Figure 5, the reward of different approaches seems to have small differences. However, the final winning rate of games varies a lot.**
>
> The gaps between the rewards and final winning rates are actually fairly similar. In Fig. 5 (left), the gaps in reward on training games (solid line) is ~0.4 between models, and the win rates for these training games are within 15% (S1-All, Table 2). The gap in reward on validation (dotted line) in Fig. 5 (left) is much greater, with G-ID and Mean-BOS getting a reward close to -1, while EMMA and O-Map obtain rewards > 0.5. This is closely mirrored by the win rates on test games (S1, Table 3), which are <20% for the G-ID and Mean-BOS baselines, but >80% for EMMA and O-Map. Similar trends between rewards and win rates are observed for experiments in Fig. 5 (middle).
>
> Perhaps there was some confusion about train/val/test rewards/win rates or stage 1 and stage 2 games? If you could point us to the specific places of confusion in the relevant figures and tables, we might be able to make some modifications to improve their clarity.
>
> **If my understanding is correct, the contribution of the paper is mainly focused on the strong generalization ability of the proposed approach across different games. However, it may need to clarify which part of the proposed approach contributes to the most significant influence.**
>
> The entity-conditioned attention mechanism is the most important component in EMMA. By extracting information from the relevant descriptor for each entity, EMMA is able to drastically simplify each task — it simply needs to learn a policy for how to interact with enemy, messenger and goal archetypes instead of memorizing a policy for each combination of entities. This facilitates knowledge sharing between games. When we ablate this attention mechanism in the Mean-BOS baseline, convergence speed is much slower, and the model overfits to the training games. We will add some discussions on why EMMA is a more effective approach compared to baselines in our revision.

---

> > ### Author Response · Authors · 2020-11-13
> > **Response (2/2)**
> >
> > [continued]
> >
> > **I am surprised to see that the EMMA approach outperforms the O-Map in the training settings.**
> >
> > By better, do you mean S2-SC in Table 2? This is the only case in which EMMA does better than O-Map, and it does so only by 1%. This is probably within the margin for noise.
> >
> > **Though utilizing the upper bound (O-Map), the winning rate still has some room to improve.**
> >
> > Even with the oracle, the hardest variants of Messenger (with a chasing enemy, fleeing message and adversarial starting position) is still a difficult control problem. The agent has to chase down the message, often by cornering it, without itself being cornered by the enemy, before bringing the message to the goal with the enemy in pursuit. It must also do this within the 64 step limit during training. The ways the entities move are provided in the text (e.g. “the mage is the enemy that is running towards you …”) to help the agent strategize accordingly. We want to emphasize again here that in RTFM, the entities all move the same way, whereas in Messenger, agents need to adapt to these variable movement dynamics.
> >
> > **I do not quite understand why the overall training process contains two rounds of stages. Is S1 stage training a warm-up training?**
> >
> > Please see the general comments on why we needed to use a curriculum (two stages). The full Messenger environment is too hard for any model to train on directly, so we pretrain on a much simpler variant. We leave learning directly on the full Messenger environment to future work.

---

### Official Review · AnonReviewer5 · 2020-11-10
**Addressing under-explored problem, but unfocused and missing clear novel contribution**

**Rating:** 6
**Confidence:** 3

**Review:**

Summary:

This paper is studying the problem of learning to interpret manuals / textual information about the task, with the goal of faster learning and generalization in RL. Unlike recent prior work (Narasimhan et al 2018, Zhong et al 2020) where the entities in the environment are already partly grounded to text (e.g. by representing them as their textual description), the agent here needs to learn the mapping between entities and corresponding text. In order to study this problem, the authors use a new environment and dataset of natural language descriptions, as well as propose a self-attention model that matches entities to relevant sentences. The proposed model performed comparable to a partly grounded policy (as in Narasimhan et al 2018).

Reasons for score:

I find the topic interesting and relevant, and the use of human-generated descriptions (in contrast to procedurally generated manuals as in Zhong et al 2020) is welcomed. However, there is relatively little in terms of novelty both in the proposed model and the environment. I think the paper would benefit from identifying and focusing on one or two of the more specific issues and exploring them in-depth.

More specifically:
(1) The proposed model is a fairly straightforward application of self-attention to text, which has been similarly used (albeit with BiLSTMs) in Zhong et al 2020.
(2) While the challenge of needing to learn which entities map onto which words is greater, each sentence refers to exactly one entity, so the agent just needs to match 3 sentences to 3 entities, which is a marginally greater challenge (if at all).
(3) With slight modifications, the environment proposed in Zhong et al 2020 should be suitable for addressing the main topic of this paper, why introduce a new environment?

Questions and comments:
- there should be a more detailed comparison to the two of the closest works (Narasimhan et al 2018, Zhong et al 2020), i.e. how exactly is your model and environment different?
- the MDP notation for transition and reward function is imprecise, actions are a random variable in both expressions (i.e. use r(s, a, s'))
- why is the curriculum introduced, is it necessary to solve the task?
- the use of the term _validation_ here is confusing, it seems you are both interested in zero-shot (on test games) and transfer learning performance (which is referred to as validation)?
- in Fig 5 right, why is learning from scratch so much worse? Shouldn't learning from scratch be as easy as in Fig 5 right/middle?

------------------------------------

UPDATE:
After reading the author's response, the reviewer's comments, and the revised version, I have increased my score.
See my response to the authors for more comments.

---

> ### Author Response · Authors · 2020-11-13
> **Response**
>
> Thank you for your comments. We hope our responses will clarify some of the novelty and scope of the challenges we address in this work that are not found in RTFM (please also see general response with detailed comparison to RTFM).
>
> **(1) The proposed model is a fairly straightforward application of self-attention to text, similar to Zhong et al. (2020).**
>
> We don’t use self-attention in our model (aside from the BERT-encoder). The query to our attention module comes from a learned entity embedding completely independent of the text (as opposed to Zhong et al (2020) which uses text to represent the entities, which are then embedded into the same space as the text manual). Our approach allows our model to select relevant descriptions and build a text-conditioned representation for *each entity*. This strategy is completely different from the one used in Zhong et al. (2020), where BiLSTM-summarized texts and global observation features modulate each other with FiLM and dot-product attention.
>
> One advantage to our approach is that the entity representations from our model can directly replace the entity embeddings typically used to embed the state observation in other models. This means that the representations from EMMA can feed directly into existing models such as txt2pi (Zhong et al. 2020).
>
> **(2) Each sentence refers to exactly one entity, so the agent just needs to match 3 sentences to 3 entities, which is a marginally greater challenge (if at all).**
>
> We explained some of the challenges in grounding entities in our environment versus RTFM in our general comments. Note that there are not always 3 descriptors; we also test the case with appended and deleted descriptions with 4 and 2 descriptors per game, respectively (Table 4). Using separate descriptions for each entity allows us to assess the overall latent mapping learned by our model (i.e. Fig. 6). But even with this assumption, Messenger is difficult: training directly on the full environment results in no learning for any model —- even EMMA cannot learn a reasonable grounding for any entity in this case (please refer to our general comments on the curriculum). This demonstrates that learning the entity grounding in our environment is a challenging problem that requires stronger entity grounding methods. We believe these remaining challenges will be of interest to the wider language grounding community.
>
> **(3) Why introduce a new environment?**
>
> For the reasons described in the general comments, we found modifying RTFM to be infeasible for answering our research questions, which are orthogonal to those RTFM is designed for.
>
> **There should be a more detailed comparison to the two of the closest works.**
>
> We clarified some of the differences between our work and those in Zhong et al. 2020 in the general comments. In Narasimhan et al. (2018), an oracle is used to concatenate the text representation to the entity representation (similar to our O-Map baseline). Access to this oracle in the wild is a strong assumption and significantly simplifies training. Our main focus is on learning a grounding without any prior mapping available. We will update the paper with more detailed discussion on both.
>
> **The MDP notation for transition and reward function is imprecise.**
>
> We’ve included the action as a subscript of the transition and reward functions. Will update based on your suggestion.
>
> **Why is the curriculum introduced?**
>
> Please see the general comments on why we needed to use a curriculum. The full Messenger environment is too hard for any model to train on directly and we leave this to future work.
>
> **The use of the term validation here is confusing, it seems you are both interested in zero-shot (on test games) and transfer learning performance (which is referred to as validation)?**
>
> You are correct - these are two different experiments.
>
> Following the standard ML train-val-test split, the validation games were used to help us design the model (and also assess generalization performance over the course of training (Fig. 5 left, middle)). The test games were used to evaluate zero-shot win rates at the end, and also feature new combinations of object movements not seen during training to assess the model’s ability to generalize to new dynamics.
>
> In a different experiment, we evaluate transfer performance by training on the training games and fine-tuning on the validation games with extra games mechanics (but the transfer itself is not the validation).
>
> **In Fig 5 right, why is learning from scratch so much worse?**
>
> In the learning curves in Fig 5 (middle), the models have been trained on an easier (stage 1) version of Messenger (Fig 5 left) as part of the curriculum. Therefore, without pre-training, learning from scratch directly in the full environment (in Fig. 5 right) is not as easy as Fig 5 middle. Furthermore, there are extra game mechanics in Fig. 5 right, namely the trap and gold items, which make the task harder.

---

### Author Response · Authors · 2020-11-13
**General Comments and Clarifications (1/2)**

We thank all the reviewers for their time and valuable feedback. We wanted to address some common questions and concerns first. We are currently revising the paper with your suggestions and will post an updated version soon.

**Clarification on single vs multi-combination games.**

In the multi-combination games, each entity can appear with every other entity. As we show with the BAM baseline, in this case, a simple co-occurrence count without acting in the environment can be surprisingly effective at recovering the underlying entity to text mapping  — there is a strong inherent bias towards the correct grounding (since, for example, the symbol for airplane is the only one that always appears together with the text “airplane”).

In contrast, for the single-combination games, we impose restrictions such that only a fixed set of entity combinations appear together during training (e.g., mage, sword, orb) (Fig. 3). Thus, any text symbol in the manual (e.g. “mage”, “enemy”, “the”, etc.) will co-occur the same number of times with all three entities, making any sort of grounding impossible without additional information. Thus, the agent must act in the environment and align the behavior of the entities with its grounding of “enemy”, “message”, “goal” in order to successfully map each entity to its corresponding text symbol. For example, if the agent has the message, interacts with entity x and obtains a reward of 1, it must infer from the description “The mage is the goal” that x must be a “mage” (assuming it has grounded “goal” correctly). This is what we mean when we say “we map symbols to text entirely through interaction with the environment”: there is no co-occurrence bias to help induce the grounding, forcing the agent to rely completely on each entity’s behavior to learn its grounding. These challenges are not found in related prior work, including RTFM (Zhong et al., 2020).

**What is the purpose of the curriculum?**

Our full Messenger environment is too difficult to solve from scratch, and none of our models can fit to even the training games. Therefore, to have a curriculum, we introduced an easier (stage 1) version of the game where all objects begin close to the agent, are immovable, and trajectories are around 4 steps to help the models learn. Grounding the entities directly with longer trajectories (up to 60 steps) and sparser reward on the full environment is very difficult and analysis of our attention map (similar to Fig. 5) shows that even EMMA cannot learn a reasonable grounding for any entity. We think this is an important challenge for future work. We mentioned this fact on page 8, but we will make sure to highlight and add more discussion on this challenge in our environment earlier in the paper.

**What is the main novelty and difference compared to RTFM (Zhong et al. 2020)?**

The main novelty of our work (both the Messenger environment and our model) is in specifically tackling the issue of *entity grounding* without any prior knowledge. To do this, Messenger (in contrast to previous work, including RTFM) (1) does not have any signal connecting entities to text symbols, (2) features much richer language, and (3) requires interaction in the environment to ground entities to text. We describe these in more detail:

1. RTFM’s observation space consists of a grid of text which shares a set of symbols with the text manual. Thus, both the text in the manual and the observation are embedded into the same space (e.g. using the same word vectors), essentially providing models with the entity grounding upfront. In contrast, our environment has a separate set of symbols for the entities with no relation to the text in our manual. Thus, the entities and text are embedded into different spaces, and learning to map between these two spaces is the key challenge in our environment that has not been explored before.
2. RTFM features only 32 total rule-based templates for the text, and each entity can only be referred to in a single way (“goblin” is always “goblin”). In contrast, we crowdsourced thousands of completely free-form descriptions in *two* rounds using Amazon Mechanical Turk. After obtaining the seed templates from the first round, we intentionally inject multiple synonyms for each entity to construct each prompt for the second round. Workers often further paraphrased these synonyms, resulting in 5, 6 or often more ways to describe the same entity (e.g. airplane, jet, flying machine, aircraft, airliner etc.). The need to map these different text symbols to the same entity further complicates the entity grounding problem in our case and more closely mirrors the challenges of grounding entities in the real world. We believe Messenger provides a much closer approximation to ‘natural’ language compared to RTFM.

---

> ### Author Response · Authors · 2020-11-13
> **General Comments and Clarifications (2/2)**
>
> [continued]
>
> 3. RTFM features all possible combinations of entities during training, which as we explained in the previous clarification on single and multi combination games, provides an additional signal that may simplify the grounding problem.
> 4. Furthermore, each entity in RTFM only moves in a single way, whereas we have different dynamics such as fleeing, chasing, and immovable entities (and this is also described in the text). This also allows us to test our model’s ability to generalize to unseen dynamics with unseen entity movement combinations, whereas in RTFM the evaluation on unseen games is essentially state-estimation.
>
> We want to emphasize that our focus is on robust grounding of entities with *rich language, variety of synonyms, in an interactive setting without any priors or co-occurrence bias to help induce this grounding*. These are all challenges orthogonal to those presented in RTFM.
>
> **Why not just extend RTFM (Zhong et al. 2020)?**
> Messenger is quite similar to RTFM and shares several aspects (e.g. grid-world with different entities, and a text manual). That said, there are numerous reasons why we were not able to adapt the original RTFM environment to meet our requirements. We enumerate them here:
> 1. The dynamics in RTFM make entity grounding (the primary focus of our work) difficult. Messenger requires much simpler reasoning than RTFM, and it is already too difficult to ground entities directly in Messenger without a handicap (see clarification on the curriculum above). RTFM sidesteps the issue by providing this grounding beforehand.
> 2. Obtaining enough crowdsourced descriptions is hard with RTFM because of the more complicated dynamics. In RTFM, there are monsters, weapons, elements, modifiers, teams, variable goals and different weaknesses between entity types that need to be specified. Collecting enough descriptions that are entirely human written would be challenging. (RTFM sidesteps this issue by using templates to generate their text manual).  In contrast, there are only entities, 3 roles, and a fixed goal in Messenger, making the text-collection task much more tractable.
> 3. The entities in our Messenger environment are carefully chosen to make entity grounding harder. In RTFM, each entity is referred to in a single way, and it’s not clear how to refer to them in multiple ways (e.g. there aren’t too many other ways to say “goblin”). In contrast, we specifically chose a set of entities that allowed for multiple ways of description, and actively encouraged this during data collection.
> 4. The combination of entities that appear during training in Messenger is carefully designed. This is so that we can introduce single-combination games and the associated grounding challenges that come with it.
> 5. We have different movement types for each entity. These different movements are referred to in our text manual and significantly increase the richness and variety of descriptions we collected, and also allow us to test generalization to unseen movement combinations. In RTFM, the entity movements are the same and fixed for all entities.
> 6. Each entity’s attribute is referenced in the observation in RTFM, e.g. the grid has entries such as “fire goblin”. We could add to the cell an extra symbol for “fire”, but this further obfuscates the entity grounding problem we are focussing on, because we would also need to obtain a grounding for all the attributes such as “fire”.
>
> We thank the reviewers for bringing up these questions and we are currently updating the paper to more clearly underscore the unique challenges we address in our work and further lay out the differences between our approach and those in Zhong et al. (2020).

---

### Author Response · Authors · 2020-11-19
**Uploaded Revised Version**

We have posted a revision (with changes in blue). Please take a look and let us know if you have further comments or questions.

Some of the changes we made were: (1) included a more in depth discussion of the novelty of our model and environment compared to previous work, (2) clarified single vs multi combination games and the purpose of the curriculum, and (3) added more discussion about the results, applications, and future work. Additional clarifications and discussion were also added to the appendix.

Thank you!

---

### Decision · Program_Chairs · 2021-01-07
**Final Decision**

**Decision:**

Reject

**Comment:**

Like the reviewers, I find this paper extremely borderline. On the one hand, it is clearly written, about a topic I find fascinating, and generally well motivated if not shockingly novel (i.e. removing some of the simplifying assumptions from Zhong et al. 2020, e.g. requiring grounding to be learned, use of real language rather than synthetically generated). On the other hand, I agree with the leitmotiv present amongst the reviews that the problem at the centre of the experimental setting is very, very simple (3 objects, 3 descriptions). I am mindful of the fact that access to computational resources is unevenly distributed, and am not expecting a paper like this to immediately scale their experiments to highly complex settings with photorealism, etc, but I can't help but feel that a more challenging task, with a deeper analysis of the problems presented by both grounding and the use of non-synthetic language, would both have been highly desirable to make this paper uncontroversially worth accepting.

As a result, the decision is to not accept the paper in its present form. Work on this topic should definitely be presented at ICLR, but it's a shame this paper did not make a stronger case for itself.